Microplastic effects on mouse colon in normal and colitis conditions: A literature review

Zolotova Natalia natashazltv@gmail.com 1
Dzhalilova Dzhuliia 1
Tsvetkov Ivan 1
Silina Maria 1
Fokichev Nikolai 1 2
Makarova Olga 1
1 Department of Immunomorphology of Inflammation, Avtsyn Research Institute of Human Morphology of Federal State Budgetary Scientific Institution “Petrovsky National Research Centre of Surgery” , Moscow , Russia
2 Faculty of Biology and Biotechnology, HSE University , Moscow , Russia
Oehlmann Jörg
Electronic publication date: 2025 Feb 10
Publication date: 2025
Volume: 13
Electronic Location ID: e18880
Received 2024 Aug 9; Accepted 2024 Dec 28
Copyright: ©2025 Zolotova et al.
Copyright year: 2025
Copyright holder: Zolotova et al.
License: This is an open access article distributed under the terms of the Creative Commons Attribution License, which permits unrestricted use, distribution, reproduction and adaptation in any medium and for any purpose provided that it is properly attributed. For attribution, the original author(s), title, publication source (PeerJ) and either DOI or URL of the article must be cited.
License URL: https://creativecommons.org/licenses/by/4.0/

Keywords: Microplastics, Colon, Colitis, Dextran sulfate sodium, Mice

Funding: Russian Science Foundation No. 24-24-00173 The work was supported by the Russian Science Foundation (No. 24-24-00173). The funders had no role in study design, data collection and analysis, decision to publish, or preparation of the manuscript.

==============================
Background

Taking into account the global spread of microplastic (MP) pollution, the problem of the MP impact on human health is relevant. MP enters the organism predominantly with water and food, and is mostly detected in the large intestine. Therefore, the connection between MP pollution and the increase in colitis is an important question. In order to assess the toxic and pathogenetic effects of MP, experimental studies were actively conducted during recent years, mainly on laboratory mice.

Objectives

The aim of our review was to summarize and systematize the data on the MP effect on mice colon under normal conditions and during colitis in order to assess the role of MP in the development of intestinal diseases. This manuscript could be relevant for ecologists, experimental biologists, and physicians dealing with problems related to anthropogenic environmental changes and inflammatory bowel diseases.

Survey Methodology

The search was conducted based on PubMed data about original experimental studies of the MP effects on the colon of healthy mice and mice with colitis.

Results

In healthy mice colon, MP can cause oxidative stress, increased permeability, immune cell infiltration, production of proinflammatory factors, and decreased mucus production. MP affects proliferation, apoptosis, and differentiation of epithelial cells, expression of tight junction components and glycocalyx, membrane transport, signaling pathways, metabolome, and intestinal microflora composition. In mice with acute and chronic experimental colitis, MP consumption leads to a more pronounced pathological process course.

Conclusions

MP may be one of the factors contributing to the development of colitis in humans. However, further research is needed.

Introduction

Microplastics (MP) includes particles up to five mm size. Moreover, nanoplastics were divided into separate category—particles ranging from 1 to 100 nm size in at least one dimension. A distinction was made between primary MP—specially created small plastic particles, such as those added to personal care products and cosmetics, and secondary MP—particles formed during the fragmentation of larger plastic products (Frias & Nash, 2019; Dube & Okuthe, 2023). Plastics are high-molecular chemical compounds obtained by industrial polymerization processes. Plastic is the key component of a wide range industrial and consumer products, including cosmetics, detergents, paints, synthetic fertilizers and pesticides. Global plastic production was steadily increasing since 1960s. In 2023, 400.3 million tons of plastic were produced worldwide. Only 9% of plastic waste is being recycled, about 19% is being burned out, and the rest ends up in landfills. In this regard, plastic pollution became a global environmental problem (Dube & Okuthe, 2023; Nayanathara Thathsarani Pilapitiya & Ratnayake, 2024). Due to its small size, MP can be easily spread by water and wind. MP particles were detected in inhaled air, in agricultural soil, in rivers, lakes, seas, oceans, drinking water, in living organisms and food products. In this regard, the issue of the MP impact on human health is relevant (Dube & Okuthe, 2023; Feng et al., 2023; Covello et al., 2024; Winiarska, Jutel & Zemelka-Wiacek, 2024).

The primary mechanism by which MP enter the human body is consumption with water and food. MP was detected in tap and bottled water, beer, milk, salt, sugar, and honey. A significant amount of MP was detected in seafood: fish, shrimp, and shellfish. In addition, MP was revealed in rice and packaged meat (Muhib et al., 2023; Zuri, Karanasiou & Lacorte, 2023; Feng et al., 2023).

There are different assessments of human MP consumption: 39,000–52,000 particles/person/year through water and food (Cox et al., 2019); up to 458,000 particles/person/year through tap water and 3,569,000 particles/person/year through bottled water (Danopoulos, Twiddy & Rotchell, 2020); over 90,000 particles/person/year (Van Raamsdonk et al., 2020); 0.1–5 g/person/week (Senathirajah et al., 2021); 2.9 × 1010 particles/person/year (Liu et al., 2021); 426 µg/kg bw/day for preschoolers (Ke et al., 2023); 96 particles/kg bw/day (average adult daily intake via food and beverages) (Zuri, Karanasiou & Lacorte, 2023).

Up to the present moment, MP particles were detected in 15 types of human biological samples, including the colon, feces, lungs, bronchoalveolar lavage, sputum, saliva, blood, liver, spleen, breast milk, placenta and meconium, hand and skin swabs and hair (Kutralam-Muniasamy et al., 2023; Barceló, Picó & Alfarhan, 2023; Yang et al., 2023b).

There are several studies indicating a link between the accumulation of MP and the development of some human diseases, but the available data are limited. Thus, as per Chen et al. (2022a), in patients with non-small cell lung cancer, MP particles were detected in tumor nodes approximately twice as frequently as in normal tissue. Horvatits et al. (2022) demonstrated that through liver cirrhosis, the MP level in this organ was higher than in samples obtained from autopsies of people without liver disease. According to Yan et al. (2022) data, the MP level in feces of patient with inflammatory bowel diseases (IBD) was higher than in feces of healthy people, and there was a positive correlation between the concentration of MP in feces and IBD severity. Cetin et al. (2023) assessed the MP level in patients with colon adenocarcinoma. The number of MP particles detected in tumor tissue was higher than in tumor-free colon areas of the same patients and in patients without colorectal cancer.

Since studies on clinical material conducting is difficult, experimental models are widely used to evaluate MP effects. The most widely used model for the toxic effects of various substances in vivo evaluation is the investigation on laboratory animals. The research on the laboratory rodent’s health MP effects began not long ago—the earliest work in the PubMed was published in 2017 (Deng et al., 2017), however by the end of 2023 there were already more than 300 studies, indicating growing interest in the scientific community to the MP impact on public health problem. Already in the first studies was demonstrated that when consumed orally, MP can cause colon epithelial barrier damage and gut microbiota composition changes, penetrate mice liver and kidneys, causing oxidative stress and disturbances in fats and carbohydrates metabolism. In addition, MP can lead to cardiomyocytes death and myocardial fibrosis development, cause cognitive and behavioral disorders, impair reproductive function and cause developmental disorders in offspring (see the details in reviews Zolotova et al., 2022; da Silva Brito et al., 2022).

When MP is consumed with water and food, the first target of its effect is the gastrointestinal tract and the highest MP concentrations in human biological samples were detected in the intestine (Kutralam-Muniasamy et al., 2023; Barceló, Picó & Alfarhan, 2023; Yang et al., 2023b). There was a suggested connection between MP consumption and the IBD development (Chen et al., 2023; Zhao, Liu & Xu, 2023; Ji et al., 2023). As noted above, two studies on human revealed a correlation between MP accumulation and bowel disorders: IBD (Yan et al., 2022) and colon adenocarcinoma (Cetin et al., 2023). IBD is a chronic and life-threatening inflammatory disease of gastroenteric tissue characterized by episodes of intestinal inflammation. The principal types of IBD are Crohn’s disease and ulcerative colitis. Crohn’s disease usually involves the terminal ileum, cecum, perianal area, and colon, but it can affect any region of the intestine in a discontinuous pattern. In contrast, ulcerative colitis involves the rectum and can affect part of the colon or the entire colon in a continuous pattern (Guan, 2019; Katsandegwaza, Horsnell & Smith, 2022; Zhao, Liu & Xu, 2023). IBD is an important risk factor for developing colorectal cancer (Guan, 2019; Shah & Itzkowitz, 2022).

According to various studies, the average content of MPs in the feces of healthy people is 2 (Schwabl et al., 2019), 28 (Yan et al., 2022), 39 (Santini et al., 2024) or 50 (Ho et al., 2022) particles per 1 g of dry weight of sample. The fecal MP concentration in IBD patients is 42 particles/g (Yan et al., 2022). There were no published data on the MPs content in the human small intestine. In human colon the mean MP content is 208–218 particles/g in control or non-tumoral tissues samples and 703 particles/g in tumoral colon tissue (Cetin et al., 2023). According to Ibrahim et al. (2021) the average count of MPs in colectomy samples was 28 particles/g. Such a large difference in the assessment of MP content is due to both different living conditions of patients and different methods of MP detection. At the present moment, there is no standardized method for analyzing the MP content in biological samples. Different researchers use various methods of sample preparation and take into account different size ranges particles.

After oral MP administration in mice the particles primarily accumulated in the intestine. Biodistribution investigation of polystyrene (PS) particles 0.05, 0.5 and 5 µm size in the mice after 24 h exposure demonstrated predominant accumulation in the intestine (Liang et al., 2021). Deng et al. (2017) investigated the accumulation of 5 µm and 20 µm PS particles in mice liver, kidney and gut. They found out that for both particle sizes tested, tissue concentration of MPs reached steady-state within 14 days of the exposure onset in all three tissues. Four weeks after the exposure, the maximal tissue concentrations of 5 µm MPs was revealed in the gut (1.4 mg/g or 2 ×107 particles/g). For 20 µm MPs distribution in organs was approximately equal (0.8 mg/g or 1.8 ×105 particles/g). Yang et al. (2019) assessed bioaccumulation and toxicokinetic/toxicodynamic of 5 or 20 µm PS particles in mice liver, kidney, and gut. They found that gut demonstrated the highest bioaccumulation factor of ∼ 8 exposed to 5 µm PS MPs with a mean residence time of ∼17 days. The mean threshold MP concentrations among the most sensitive biomarkers were 8 µg/g bw for 5 µm particles and 0.71 µg/g bw for 20 µm particles. Therefore, experimental studies of the MP’s effect on the intestine in healthy and in intestinal diseases conditions are relevant.

The small and large intestines differ according to many characteristics: morphological structure, cellular composition, gene expression, functioning, microbiome, immune responses, diseases, etc (Bowcutt et al., 2014; Jensen et al., 2023). Therefore, these two parts of the intestine should be investigated separately. Currently, most researches of MP effects on the gut are focused on the colon. In PubMed throughout all the time up to 04/2024 search for (“microplastics” AND (“large intestine” OR “colon” OR “large bowel”)) yields 57 articles, when search for (“microplastics” AND (“small intestine” OR “small bowel” OR “duodenum” OR “jejunum” OR “ileum”)) yields only 29 articles. That is why we decided to focus our review to the MP effects on the mice colon.

Since there is evidence linking MPs to human IBD and human colorectal cancer, we examined experimental studies on this subject. Numerous mouse models of IBD have been developed, including chemically induced, cell-transfer, congenial mutant, and genetically engineered models. Chemically induced models, mainly dextran sodium sulfate (DSS)- and 2,4,6-trinitrobenzenesulfonic acid (TNBS)-induced colitis, are most commonly used (Mizoguchi, 2012; Gadaleta, Garcia-Irigoyen & Moschetta, 2017; Katsandegwaza, Horsnell & Smith, 2022). Search for (“microplastics” AND “mice” AND (“inflammatory bowel disease” OR “colitis” OR “Crohn’s disease”)) throughout all the time up to 04/2024 revealed studies only with the DSS-induced colitis model. Search for (“microplastics” AND “mice” AND “TNBS”) did not reveal any results. Search for (“microplastics” AND “mice” AND (“colon cancer” OR “colorectal cancer”)) revealed only one experimental study. Yang et al. (2023a) demonstrated in a transplant model of colorectal cancer (CT26-Luc cells were injected near the colon) that polyethylene and polystyrene particles of about 0.5 µm size can promote tumor initiation and development. In addition to IBD and tumors, common diseases of the colon include infectious colitis, diverticulosis and irritable bowel syndrome, but there was no data on their connection with MPs in either humans or mice. Therefore, we focused our attention on the MPs effects on the course of DSS-induced colitis in mice.

We hope that our review will be practical and useful for ecologists, experimental biologists, and physicians dealing with problems related to anthropogenic environmental changes and inflammatory bowel diseases.

Survey Methodology

To assess the MP effect on the healthy mice colon, we examined the PubMed for those articles included queries “microplastics, colon, mice”, “microplastics, large intestine, mice” and “microplastics, large bowel, mice” through all the time up to 04/2024. Totally 60 publications were found and then 34 materials were selected. Inclusion criteria were: (1) original experimental research; (2) at least part of the work was performed on mice in vivo, (3) MP was added to the animals’ water, food, or administered by gastric intubation, (4) changes in the colon wall and/or intestinal microflora under the influence of MP were studied, (5) the full text of the article is open access. Exclusion criteria: (1) reviews, protocols, preprints, corrections to the article (2) the work was carried out entirely on cell cultures or human biological samples, (3) MP was administered intratracheally, (4) no access to the full article text, (5) the effects of the MP on the colon are not mentioned, (6) the article describes combined MP effect with other factors (metals, probiotics, drugs, etc.) in case it was not possible to isolate the MP itself effects (Fig. 1).

To assess the MP effect on the course of the inflammatory process in the colon, we searched PubMed and Google for those articles included queries “microplastics, colitis, mice”, “microplastics, Inflammatory bowel disease, mice”, “microplastics, dextran sulphate sodium, mice”, “microplastics, DSS, mice” for all time up to 04/2024. Inclusion criteria were: (1) original experimental research; (2) at least part of the work was performed on mice in vivo, (3) MP was added to animals’ water, food or administered by gastric intubation; (4) animals that demonstrated inflammation of the colon before, during, or after MP consumption. Exclusion criteria: (1) reviews, protocols, preprints, corrections to the article; (2) the work was carried out entirely on cell cultures or human biological samples; (3) MP was administered by intratracheal instillation. Only seven works were found that were in accordance of the given criteria (Fig. 2).

Figure 1 PRISMA flow for MP effect on the healthy mice colon.

Figure 2 PRISMA flow for MP effect on the course of colitis.

Results

Effect of MP on mouse colon in normal conditions (healthy animals)

The experimental conditions in the analyzed articles varied considerably (Table 1). Animal sex: 21 studies were performed on males, four on females, three on animals of both sexes, and six studies did not specify sex. Animal strain: 24 studies were performed on C57BL/6, C57BL/6J, or C57BL/6N mice, six studies on ICR, two on BALB/c, and one each on Kunming and C57BKS/Leprdb. Type of plastic: 26 used polystyrene (PS) particles, four studies used polyethylene (PE), single articles used polypropylene (PP), polyvinyl chloride (PVH), polyethylene terephthalate (PET), and low-density polyethylene (LDPE), two studies used biodegradable plastics: PLGA (poly lactic-co-glycolic acid) and PLA (polylactic acid), and one study did not specify the plastic type. Particle size: six studies used nanoplastics (particle diameter from 20 to 100 nm). MP particle size ranged from 0.14 to 300 µm, with particles with a diameter of 0.5 µm (seven articles) and 5 µm (11 articles) was the most commonly used. Dosages: ranged from 0.002 to 500 mg MP/kg of body weight per day, with dosages from 0.2 to 20 mg/kg/day mainly used (18 articles). Exposure: varied from one day to three months, mostly 4–6 weeks.

Table 1 MP effects on the healthy mice colon.

References	Experimental conditions	Changes, detected in the colon during MP consumption	
Chen et al. (2022b)	Male C57BL/6J mice
PS 5 μ m
dose 7.5 mg/kg/day (probe 0.1 mg/mice/day)
consumption duration 42 day	Intestinal microbiota changes
Metabolic changes	
Chen et al. (2022c)	Male C57BL/6 mice
PVC 2 μ m 100 mg/kg/day (probe)
consumption duration 60 day	↓ Goblet cells number
↓Muc2, Klf4, Retnlb mRNA expression
↓Muc1 and Muc3 mRNA expression
↑ Dextran intestinal permeability
Intestinal microbiota changes
Metabolic changes	
Choi et al. (2021a)	ICR mice
PS 0.5 μ m
dose approximately 0.2, 1 and 2 mg/kg/day (probe 10 μ g/ml, 50 μ g/ml or 100 μ g/ml on 0.5 ml/day)
consumption duration 14 days	↓ Mucosal thickness, muscle thickness and surface epithelium thickness
↓ Goblet cells number
↑Tnfa, Tgfb, Il1b, Il6 and Il10 mRNA expression
ASC-inflammosome signaling pathway:
↑ NLRP3, ASC levels, split caspase 1/ caspase 1 ratio
NF-κB signaling pathway:
↑ p-NF-κB, p-I κB-α, mRNA Nfkb levels	
Choi et al. (2021b)	ICR mice
PS 0.5 μ m
dose approximately 0.2, 1 and 2 mg/kg/day (probe 10 μ g/ml, 50 μ g/ml or 100 μ g/ml on 0.5 ml/day)
consumption duration 14 days	↓ Whole intestine length (middle and high MP doses)
↑ Crypts length
↓ Thicknesses of mucosa, thicknesses of muscle, thicknesses of flat luminal surface, thicknesses of crypts layer
↓ Goblet cells number
↓ Acid mucins of the Goblet cells level
↓Muc2 and Klf4 mRNA expression irregular in size and shape Goblet cells, average number of mucin vacuoles in each of the Goblet cells was significantly increased
↓ Cholecystokinin (all doses) and gastrin (medium and high doses) levels
↓Muc1 mRNA expression
↓ Cl− dose dependent level
↓ Chloride channel Clc2 and Cftr mRNA expression
↓ Water channels Aqp3 and Aqp8 mRNA expression
muscarinic acetylcholine receptor signalling pathway:
↓ mAChR M2 and mAChR M3 levels, however
↑ Gα protein level and
↑ Phosphorylation level
PKC and PI3K (excluding low dose)
MAPK/NF-κB signalling pathway:
↑ ERK, p38, NF-κB and IκB phosphorylation levels	
Djouina et al. (2022)	Female C57BL/6 mice
PE 36 and 116 μ m
dose approximately 16.6 mg/kg/day (100 μ g MP/g of food)
consumption duration 42 day	↑ Crypts depth (MP mix)
↑ Stem cell marker intestinal Lgr5 mRNA expression (large MP particles)
↑ Goblet cells volume (mucin-positive area) (MP mix)
↑Muc2 mRNA expression (MP mix)
↑Vil1 mRNA expression (large particles and mix)
↑Chga mRNA expression (large particles)
↑Ocln, F11r mRNA expression (large particles and mix) (large particles and mix)
↑ Polymorphonuclear neutrophils (large particles)
↓ Anti-inflammatory macrophages (large particles and mix)
no changes: Tnfa, Il1b mRNA expression
↑Ifng mRNA expression (large particles and mix)
mRNA expression
↑Il6 mRNA expression (MP mix), no changes: intestinal microbiota	
Garcia et al. (2024)	Male and female C57BL/6 mice
PS 5 μ m, or mix plastics (5 μ m) PS, PP and PLGA
single doses 45 and 90 mg/kg, twice a week (2 and 4 mg/mice/week through probe)
consumption duration 28 days	Metabolic changes	
Huang et al. (2023)	Female C57BL/6 mice
PS 5 different size: 0.02, 0.05, 0.1, 0.2 and 0.5 μ m
dose 75 mg/kg once a 2 day
consumption duration 42 day	Infiltration (PS-500 and PS-200)
↑ Intervals between crypts, (PS-500 and PS-200)
↓ Goblet cells number (PS-500 and PS-200)
↑ Monocytes (CD11b+Ly6C+) (PS-500)
↑ Neutrophils (CD11b+Ly6G+) (PS-500)	
Jia et al. (2023)	Male C57BL/6 mice
PP 8 and 70 μ m
doses 1, 10 and 100 mg/kg/day (probe)
consumption duration 28 days	Inflammatory infiltrate by lymphocytes (1,0 and 10 mg/ml PP size of 8 μ m), edema submucosa (all experimental groups), glandular mucosa hyperplasia (10 mg/ml PP size 70 μ m). Mitochondrial disorders,
↑ Died epithelial cells number (all experimental group),
↑ Bax level (all groups),
↓ Bcl-2 level (all groups),
↑ Cleaved caspase-9 levels (all groups) and cleaved caspase-3 (low and high doses)
↓ Goblet cells number (mid and high doses)
↓ Muc1 level,
↓ Cldn1 (8 μ m mid and high doses and 70 μ m high dose) occludin (all groups), ZO-1 levels (8 μ m mid and high doses dose and 70 μ m high dose),
↓ NKCC1+ cells volume fraction (all experimental group),
↓ SLC26A6+ cells volume fraction (8 μ m all doses and 70 μ m mid and high doses),
↓ CFTR+ cells volume fraction (high doses both size).
↑ MDA level (all groups),
↓ GSH level (all groups),
↑ GSSG (oxidized glutathione) level (8 μ m and 70 μ m concentration at 1 mg/ml and 10 mg/ml),
↓ SOD, GPx, CAT levels (all groups),
↑ TNF-α, IL-1β and IL-6 levels
↓ IL-10 level,
TLR4/NF-κB signalling pathway: ↑ TLR4, p50 and p-p65levels, ↓ IκB level	
Jiang et al. (2024)	Male and female C57BL/6J mice
PS 0.5 μ m
dose approximately 0.45 and 4.5 mg/kg/in 2 days (probe 0.01 mg or 0.1 mg on mice once in 2 days)
consumption duration 35 days	↑ 4 kDa dextran intestinal permeability (high dose group)
Intestinal microbiota changes
Metabolic changes	
Li et al. (2020)	Male C57BL/6 mice
PE 10–150 μ m
doses 0.3, 3 and 30 mg/kg/day (in food 6, 60 and 600 μ g/day)
consumption duration 35 days	Lamina propria inflammatory infiltrate by lymphocytes and plasma cells (dose 600 μ g/day).
↑ Crypts length (dose 600 μ g/day), edema lamina propria (dose 600 μ g/day)
↑ Histologic score (dose 600 μ g/day)
TLR4 signaling pathway:
↑ TLR4, AP-1, IRF5 levels (high dose).
Intestinal microbiota changes	
Liang et al. (2021)	Male and female C57BL/6 J mice
PS 0.05 and 0.5 μ m; doses 250 or 500 mg/kg/day (probe) 1 day
PS 0.05 and 0.5 μ m; doses 2.5, 25, 50, 250 or 500 mg/kg/day (probe) 28 days	Morphological alterations were not detected (1 day)
↑ Died epithelial cells number (1 day all doses and size; 28 day PS50 and PS500 doses from 50 mg/kg and higher)
↓ Goblet cells number (1 day, 0,05 μ m high dose, 0,5 μ m both doses)
↓Muc2 mRNA expression (1 day)
↓Ecad mRNA expression (1 day, 0,05 μ m high dose, 0,5 μ m both doses)
↓Muc1, Muc3, Muc13 mRNA expression (1 day)
↑Cldn3, Cldn4, Cldn7, Ocdn mRNA expression (1 day, mix and high doses)
↑ Intestinal permeability (1 and 28 day)
↑ ROS levels (1 day and 28 day, all doses and sizes)	
Liang et al. (2024)	Male C57BL/6 mice
PS 2 μ m
doses 0.5 and 2 mg/kg/day
consumption duration 56 days	↑ Intestinal permeability (both doses),
↑ LPS blood serum level (dose 2 mg/kg)	
Liu et al. (2022a)	Male C57BL/6 mice
PS 5 μ m
dose 0.125 mg/kg/day (in the water 500 μ g/l)
consumption duration 28 days	Slight vacuolization of mucosa,
Loose Goblet cells structure
↓Cldn1 and Ocln1 mRNA expression
↑ Glutathione peroxidase (GPx) mRNA expression
↑ TNF-α, IFN-γ, IL-1β levels
Intestinal microbiota changes.
Metabolic changes	
Liu et al. (2022b)	Mice C57BKS/Leprdb
PS 0.1 and 5 μ m
dose 0.05 mg/kg/day (in the water 200 μ g/l)
consumption duration 28 days	Intestinal microbiota changes	
Luo et al. (2022)	Male C57BL/6 mice
PS 5 μ m
dose 20 and 200 mg/kg/day (probe with 0.5 or 5 μ g in 200 μ L water) 21 days
dose 20 mg/kg/day (probe with 0.5 in 200 μ L water) 63 days	↑Tnfa mRNA expression (21 day)
↑Il17a mRNA expression (21 day)
↑Il22 mRNA expression (63 day)	
Lv et al. (2023)	Male C57BL/6 mice
PS 5 μ m
dose 0.04 mg/kg/day (in the water 200 μ g/l)
consumption duration 35 days	↑ DAO, D-Lac, IFABP and D-LDH levels in blood serum
Intestinal microbiota changes	
Park et al. (2023)	Male C57BL/6 mice
(unknown type of MP) 0.05 μ m
dose unknow
consumption duration 7 days	No changes: Colon weight
↑ Dying epithelial cells number
↑ Neutral mucin level
No changes: acid mucin level	
Rawle et al. (2022)	Female C57BL/6J mice
PS 1 μ m
dose 0.08 mg/kg/day (in the water 526 μ g/l)
consumption duration 33 day	RNA seq demonstrated ↑ mitochondrial activity and ↑ ROS activity
RNA seq indicated moderate ↑ proinflammatory Signal pathway, ↓ signalling pathway sirtuins, ↑ Ribosomal and translational activity, genes related to cytoskeleton and interaction with extracellular matrix expression changes
MP consumption had no significant effect on the intestinal microbiome	
Shaoyong et al. (2023)	Male C57BL/6 mice
PS 0.14 μ m
dose 0.25 mg/kg/day (probe)
28 days, then 3 days without MP	↓ Colon length
↓ Goblet cells number
↓Muc2 levels (WB)
↓ Muc1 levels (WB)
↓ Ocdn1, ZO-1, Catenin beta-1 levels
↑ 4 kDa dextran intestinal permeability
↑ DAO, D-Lac serum levels
↑ Bacterial translocation to spleen and liver
↑ ROS and N2O2 levels
No changes: GSH level
↓ SOD2 level
No changes: GPx4 level
↓ CAT level
No changes: T-AOC (total antioxidant capacity)
↑ TNF-α and IL-1β levels	
Shi et al. (2022)	Male ICR mice
PS 1 μ m
dose approximately 1.8 mg/kg/day (in the water 10 mg/l; dose 55 μ g/day)
consumption duration 7 and 14 days	Inflammatory infiltration submucosa, submucosa edema,
Goblet cells number reduction
Intestinal microbiota changes
Metabolic changes	
Sun et al. (2021)	Female ICR mice
PE 1–10 μ m
dose 0.002 and 0.2 mg/kg/day (probe)
consumption duration 30 days (feces were obtained at 15th day)	There were no pathomorphological changes
↓ mucin density (only high dose - 0.2 μ g/g/d group)
Muc2 mRNA expression had tendency to ↑
↓Il1b mRNA expression
No changes: IL-6 mRNA expression
↑Il8 mRNA expression
↑Il10 mRNA expression
TLR4 signalling pathway: ↓ mRNA expression Erk1, Nf-κb, however Tlr4, MyD88 not changed
No changes: short-chain fatty acid receptors
No changes: mRNA Gpr41 and Gpr43
Intestinal microbiota changes
Metabolic changes	
Sun et al. (2024)	Male C57BL/6J mice
PS 0.5 μ m
dose 5 mg/kg/day (probe)
consumption duration 30 days	↑ Mass and size of the cecum
Inflammatory infiltration, increased distance between crypts
↑ Histologic score
↓ Goblet cells number (stain AB-PAS)
↓ Claudins-1, Occludin, ZO-1 (protein) levels
↑ intestinal permeability
↑ bacterial translocation to mesenteric lymph nodes liver and spleen
↑Tnfa and Il1b mRNA expression
No changes: Il6 mRNA expression
Intestinal microbiota changes	
Wang et al. (2023a)	Male ICR mice
PS 1 μ m
dose 1 mg/kg (probe) 3 time in week
consumption duration 5 weeks	Inflammatory infiltrate
↓ Goblet cells number
↓ Claudins -1 level
No changes: ZO-1 levels
↑ TNF-α and IL-1 levels
Intestinal microbiota changes
Metabolic changes	
Wang et al. (2023b)	Male BALB/c mice
biodegradable plastic PLA (polylactic acid) nanoplastics and PLA oligomers (unknow size)
dose approximately 0.5, 5 and 50 mg/kg/day consumption duration 7 days	 Inflammatory infiltrate
(PLA in the dose of 0,01 mg/day/mice)
↑ Histologic score
↓ Goblet cells volume
↑ TNF-α level	
Wang et al. (2023c)	Male C57BL/6 mice
low-density polyethylene (LDPE) and oxidized low-density polyethylene (Ox-LDPE) from 2.67 to 12.61 μ m
dose approximately 50 mg/kg/day (probe 5 mg/ml 200 μ l)
consumption duration 28 day	 Inflammation limited to mucosa
↑ Crypts depth
↑ MDA levels,
↓ GSH levels,
↓ SOD activities
↑Tnfa, Il1b and Il6 mRNA expression
Intestinal microbiota changes
Metabolic changes	
Wen et al. (2022)	Male C57BL/6 mice
PS 5 μ m
doses 0.018 and 0.18 mg/kg/day (dissolve in water 100 μ g/l and 1,000 μ g/l) consumption duration 90 days	 ↓ Colon length (high dose)
Inflammatory infiltration, increased distance between crypts,
↓ Crypts depth (high dose)
↓ Muscle thickness (both doses)
↑ Histologic score
↓ Goblet cells number (both doses)
↓ Muc2 mRNA expression (high dose)
↓ Cldn1, Ocln, ZO-1 mRNA expression
↑ Intestinal permeability (high dose)
↑ MDA levels (high dose)
↓ GSH levels (high dose),
↓ SOD activities (high dose),
↑ TNF-α and IL-6 levels (high dose)
↓ IL-10 levels (high dose)
Intestinal microbiota changes.
Metabolic changes	
Xie et al. (2022)	Kunming mice
PE, PP, PS, PVC and PET, average diameter 150–300 μ m
dose approximately 120 mg/kg/day (probe 20 mg/ml once a day 0.2 ml/day)
consumption duration 7 days	Inflammatory infiltrate by lymphocytes,
↓ Goblet cells number
↑ MDA levels (PP, PS, PVC, PE, PET)
↑ GSH levels (PP, PS, PVC, PET)
↑ SOD levels (PP, PS, PVC, PE)
↑ POD levels (PP, PS, PVC, PE, PET)
Inflammatory cells in mucosa number: PS>PVC>PET>PE>PP>control. PS - 21,53% from all the cells; PP- 2,79%, control group - 1,47%.
Intestinal microbiota changes	
Xie et al. (2023)	Male C57BL/6mice
PS 5 μ m
dose 0.024 mg/kg/day (in the water 100 μ g/l)
consumption duration 42 days	↑ Crypts number
↑ Crypts depth
↑ Stem cell marker, intestinal Lgr5, Bmi1 and Olfm4mRNA expression
↑ Volume fraction of proliferating c-Myc and PCNA positive cells
↓ Goblet cells number
↓Tff3 and Muc2 mRNA expression
↑Il1b and Il6 mRNA expression
Notch pathway:
↑Dll1, Dll4, Jag1, Hes1 mRNA expression	
Xu et al. (2021)	BALB/c mice
PS, PS-COOH and PS-NH2 0.1 μ m
dose approximately 40 mg/kg/day (probe 1 mg/day once a day)
consumption duration 28 days	↓ Colon weight (PS-NH2 exposure)
Inflammatory infiltrate, epithelial damage, crypts dysplasia	
Yu et al. (2024)	Male mice C57BL/6N
PS 0.1 μ m
dose 5 mg/kg/day (probe)
consumption duration 28 days	↑ Goblet cells volume
No changes: SOD activities
No changes: GPx activities
Intestinal microbiota changes	
Zeng et al. (2024)	C57BL/6 J mice
PS 0.2; 1 or 5 μ m
dose 1 mg/kg/day (probe)
consumption duration 28 days	 Inflammatory infiltrate (PS5 >PS0.2 and PS1)
↓ Goblet cells number (stain AB-PAS) (PS5),
↓Muc2 mRNA expression (all sizes MP)
↓Muc1 mRNA expression (PS5)
↓Cldn1 (PS5), Ocln (PS5), ZO-1 mRNA expression and protein levels (all sizes MP)
↑ DAO and D-Lac blood serum levels (PS5)
↑ MDA levels (PS5)
↑ 3-nitrotyrosine (PS5) levels
↓ SOD activities (PS5)
↓ CAT activities (PS5; PS1)
↑Tnfα, Il1b and Il6 mRNA expression (PS5),
↑Il8 mRNA expression (PS5, PS1)	
Zha et al. (2024)	Male ICR mice
PS 5 μ m in dose 22 mg/kg/day (probe 500 μ g/mice/day)
PS 0.099 μ m in dose 7 and 22 mg/kg/day (probe 200 and 500 μ g/mice/day)
consumption duration 35 days	Morphological alterations were not detected
Intestinal microbiota changes
Metabolic changes	
Zhao et al. (2022)	Male C57BL/6 mice
PS 0,5 μ m
doses 0.18 mg/kg/day (in the water 1 μ g/ml) consumption duration 12 week	Intestinal microbiota changes were detected	
Zolotova et al. (2023)	Male C57BL/6 mice
PS 5 μ m
dose 2.3 mg/kg/day (in the water 10 mg/ml)
consumption duration 42 days	Morphological alterations were not detected
↓ Goblet cells number
↑ Sulfated mucins in the Goblet cells increased levels,
No changes: neutral mucins in the Goblet cells levels
↑ Chromogranin A+ endocrine cells number
↑ Lamina propria cells number	
Notes.

MP microplastic

LDPE low-density polyethylene

Ox-LDPE oxidized low-density polyethylene

PE polyethylene

PET polyethylene terephthalate

PLA polylactic acid

PLGA poly lactic-co-glycolic acid

PP polypropylene

PS polystyrene

PVH polyvinyl chloride

AB-PAS alcian blue and Periodic acid–Schiff staining

Macroscopic changes in the colon of mice exposed to MP were described in six studies. MP caused the colon shortening (Wen et al., 2022; Shaoyong et al., 2023) or the entire intestine (Choi et al., 2021b). According to Xu et al. (2021) colon mass decreased under the MP exposure, while according to Park et al. (2023) it did not change. In article of Sun et al. (2021), MP consumption led to a significant increase in the length and mass of the cecum.

Most of the analyzed studies (22 out of 34) presented the colon sections stained with hematoxylin and eosin histological examination. At a qualitative level, according to three studies (Sun et al., 2021; Zolotova et al., 2023; Zha et al., 2024) MP did not cause obvious pathological or inflammatory changes. According to 11 studies, MP consumption led to colon mucous membrane and submucosa infiltration by immune cells (Li et al., 2020; Shi et al., 2022; Wen et al., 2022; Xie et al., 2022; Jia et al., 2023; Wang et al., 2023b; Wang et al., 2023a; Huang et al., 2023; Zeng et al., 2024; Sun et al., 2024). A number of studies described the increase in the distance between crypts (Li et al., 2020; Choi et al., 2021b; Wen et al., 2022; Huang et al., 2023; Sun et al., 2024) and crypt dysplasia (Xu et al., 2021). Also noted the decrease in the number of goblet cells (Choi et al., 2021a; Shi et al., 2022; Xie et al., 2022; Huang et al., 2023), edema of the submucosa or mucous membrane lamina propria (Li et al., 2020; Shi et al., 2022; Jia et al., 2023), glandular hyperplasia of the mucous membrane (Jia et al., 2023), damage of the mucus layer (Wang et al., 2023c), destruction of the epithelium (Xu et al., 2021). During the morphometric study, same scientists (Djouina et al., 2022; Xie et al., 2022; Xie et al., 2023; Wang et al., 2023c) revealed an increase in the depth of the crypts, while other (Choi et al., 2021b; Choi et al., 2021a; Wen et al., 2022) on the contrary, detected the decrease in the crypts length and the mucous and muscularis externa thickness. In addition, Choi et al. (2021a), Choi et al. (2021b) noted a decrease in the superficial enterocytes height, while Xie et al. (2023) detected the increase in the number of crypts. In four studies, the histopathological score was calculated, assessing semi-quantitatively the severity of inflammatory infiltration, the prevalence and depth of damage, crypt damage, the edema absence or presence. In all experiments, the researchers investigated a statistically significant increase in the Histology score when exposed to MP (Li et al., 2020; Wen et al., 2022; Wang et al., 2023b; Sun et al., 2024).

Much attention was paid to MP effect on the colon epithelium. MP enhanced colon epithelial cells apoptosis and proliferation. The mRNA expression of intestinal stem cell markers increased: Lgr5, Bmi1 and Olfm4, and the volume fraction of proliferation markers c-Myc and Pcna in the colon also increased (Djouina et al., 2022; Xie et al., 2023). MP caused an increase in the number of dying TUNEL+ cells, the level of cleaved caspase-9, cleaved caspase-3 and the pro-apoptotic factor Bax (Liang et al., 2021; Jia et al., 2023; Park et al., 2023). The level of the anti-apoptotic factor Bcl-2 in the colon under the MP influence significantly decreased (Jia et al., 2023).

Most often, the morphofunctional state of goblet cells and the mucus they produced was assessed (17 out of 34 articles). According to the majority of authors, MP caused a decrease in the goblet cells number and volume fraction (Choi et al., 2021b; Liang et al., 2021; Sun et al., 2021; Choi et al., 2021a; Chen et al., 2022c; Wen et al., 2022; Xie et al., 2023; Jia et al., 2023; Wang et al., 2023b; Zolotova et al., 2023; Shaoyong et al., 2023; Wang et al., 2023a; Zeng et al., 2024; Sun et al., 2024). However, Djouina et al. (2022) and Yu et al. (2024) reported an increase in the goblet cells volume fraction. Ultrastructural study revealed that after MP exposure, goblet cells were inconsistent in shape and uneven in size. The average number of mucus drops in each goblet cell was significantly increased (Choi et al., 2021b). The main structural component of mucus produced by goblet cells is the mucin Muc2. According to the most studies, the glycoprotein Muc2 level (Shaoyong et al., 2023) and the expression of its mRNA (Choi et al., 2021b; Liang et al., 2021; Chen et al., 2022c; Wen et al., 2022; Xie et al., 2023; Zeng et al., 2024) in the colon decreased under the MP influence. However, Djouina et al. (2022) revealed an increase in the Muc2 mRNA expression. It was also demonstrated that MP could cause a decrease in other mucus components mRNA expression: Tff3 (Trefoil factor 3) (Xie et al., 2023), Klf4 (Kruppel-like factor 4) (Choi et al., 2021b; Chen et al., 2022c), Retnlb (Resistin-like beta) (Chen et al., 2022c). The terminal carbohydrate groups of secretory mucins could be modified by sulfuric or sialic acid residues (acidic mucins) or unmodified (neutral mucins). There were contradictory data on changes in the acidic and neutral mucins in goblet cells level when exposed to MP. Acidic mucins were stained with alcian blue. According to Zolotova et al. (2023) the intensity of Alcian blue staining increased when exposed to MP, according to Park et al. (2023)—it did not change, and according to Choi et al. (2021b)—it decreased. The intensity of the PAS reaction, which detects neutral mucins, did not change when exposed to MP according to Zolotova et al. (2023) and Park et al. (2023)—it increased.

There were few data on MP effect on other types of epithelial cells. According to Liang et al. (2021) the expression of Ecad mRNA, an epithelial cell marker E-cadherin, was reduced in mice treated with MP. Djouina et al. (2022) demonstrated that exposure to MP increased the mRNA expression for the absorptive epithelial cell marker villin 1 (Vil1) and the enteroendocrine cell marker chromogranin A (Chga). We also previously investigated the increase in the chromogranin A+ endocrine cells number in the colonic mucosa when exposed to MP (Zolotova et al., 2023). Choi et al. (2021b) reported a decrease in the cholecystokinin level and gastrin hormones in the colon of mice consuming MP.

MP affected the expression of transmembrane mucins that were part of the colon glycocalyx. According to the most studies, the Muc1 mRNA glycoprotein level (Choi et al., 2021b; Liang et al., 2021; Chen et al., 2022c; Jia et al., 2023; Shaoyong et al., 2023; Zeng et al., 2024), mRNA Muc3 (Liang et al., 2021; Chen et al., 2022c) and mRNA Muc13 (Liang et al., 2021) decreased under the MP influence.

Much attention was paid to MP effect on tight junctions in the colonic epithelium. The claudins Cldn2, Cldn3, Cldn4, Cldn7 mRNA expression in the colon increased when exposed to MP (Liang et al., 2021). In contrast, the expression of mRNA and protein level of claudin Cldn1 decreased (Liu et al., 2022a; Wen et al., 2022; Jia et al., 2023; Wang et al., 2023a; Zeng et al., 2024; Sun et al., 2024). According to Liang et al. (2021) and Djouina et al. (2022) the occludin mRNA expression increased, and according to other works (Liu et al., 2022a; Wen et al., 2022; Jia et al., 2023; Shaoyong et al., 2023; Zeng et al., 2024; Sun et al., 2024) both the expression of mRNA and the occludin protein level decreased. The mRNA expression and the ZO-1 protein level decreased (Wen et al., 2022; Jia et al., 2023; Shaoyong et al., 2023; Wang et al., 2023a; Zeng et al., 2024; Sun et al., 2024). According to Djouina et al. (2022) the expression of F11r mRNA (junctional adhesion molecule A) increased, and according to Shaoyong et al. (2023) the β-catenin protein level decreased.

Two studies assessed the membrane transport of intestinal epithelial cells. When exposed to MP Jia et al. (2023) revealed a decrease in the NKCC1 (Na+-K+-2Cl− cotransporter 1), SLC26A6 (solute carrier family 26 member 6) and CFTR (cystic fibrosis transmembrane conductance regulator), level, and Choi et al. (2021b) revealed a decrease in the level of chloride ions, a decrease in the chloride channels Cftr, Clc2 and aquaporins Aqp3, Aqp8 mRNA expression.

MP led to the increased intestinal permeability. A number of studies demonstrated increased intestinal permeability for fluorochrome-labeled dextran with a molecular weight of 4 kDa and 70 kDa (Liang et al., 2021; Chen et al., 2022c; Wen et al., 2022; Shaoyong et al., 2023; Liang et al., 2024; Jiang et al., 2024; Sun et al., 2024). Increased intestinal permeability with MP consumption was also evidenced by the increased diamine oxidase (DAO), D-Lactate, intestinal fatty acid-binding protein (IFABP), D-Lactate dehydrogenase (Lv et al., 2023) and lipopolysaccharide (LPS) (Liang et al., 2024) blood levels (Shaoyong et al., 2023; Lv et al., 2023; Zeng et al., 2024). In addition, bacterial translocation to the mesenteric lymph nodes, liver, and spleen increased (Shaoyong et al., 2023; Sun et al., 2024).

MP induced oxidative stress. In mice treated with MP, the production of reactive oxygen species (ROS) was increased in the colon (Liang et al., 2021; Shaoyong et al., 2023). RNA seq data also indicated the increase in mitochondrial activity and ROS production (Rawle et al., 2022). The lipid peroxidation marker malondialdehyde (MDA) level increased (Wen et al., 2022; Xie et al., 2022; Jia et al., 2023; Wang et al., 2023c; Zeng et al., 2024) and the tyrosine oxidation product 3-nitrotyrosine (Zeng et al., 2024), and the level of an important intracellular antioxidant, glutathione (GSH) levels decreased (Wen et al., 2022; Jia et al., 2023; Wang et al., 2023c), but the level of its oxidized form GSSG (Glutathione disulfide) increased (Jia et al., 2023). However, in the Xie et al. (2022) study, the GSH level in the colon increased under the MP influence, while in the work of Shaoyong et al. (2023) it was not described as changed, in addition, the total antioxidant capacity did not change. Furthermore, MP affected the production and antioxidant enzymes activity such as superoxide dismutase (SOD), glutathione peroxidase (GPx), catalase (CAT), and peroxidase (POD). According to the provided data, the level and SOD activity decreased under the MP influence (Wen et al., 2022; Jia et al., 2023; Shaoyong et al., 2023; Wang et al., 2023c; Zeng et al., 2024), however, in the work of Yu et al. (2024) SOD activity did not change, and in the study of Xie et al. (2022) SOD level increased. Regarding GPx, the obtained data were contradictory: it was reported the increase in GPx mRNA expression, Jia et al. (2023) claimed the decrease in GPx level, and Shaoyong et al. (2023) and Yu et al. (2024) did not reveal changes in the level and activity of this enzyme. The colon CAT level and activity decreased (Jia et al., 2023; Shaoyong et al., 2023; Zeng et al., 2024). One study noted an increase in peroxidase activity (Xie et al., 2022).

As mentioned above, most studies qualitatively noted colon mucosa immune cell infiltration when exposed to MP. In four studies, infiltration was assessed quantitatively. According to our data (Zolotova et al., 2023) MP caused a statistically significant increase in the number of cells in the lamina propria of the mucosa. Djouina et al. (2022) demonstrated the increase in the ratio of polymorphonuclear neutrophils and a decrease in the ratio of anti-inflammatory macrophages. In the work of Huang et al. (2023) an increase in the CD11b+Ly6C+ proinflammatory monocytes and CD11b+Ly6G+ neutrophils ratio was observed. Xie et al. (2022) identified the ratio of inflammatory cells in the colon mucosa depending on the type of MP used: PS > PVC > PET > PE > PP > control (when using PS, inflammatory cells accounted for 21.53% of all cells; PP—2.79%, in the control group—1.47%).

MP affected the cytokines in the colon production. According to most studies, MP exposure increased the proinflammatory cytokines TNFα, IL-1β and IL-6 level, as well as their mRNA expression (Choi et al., 2021a; Liu et al., 2022a; Djouina et al., 2022; Wen et al., 2022; Luo et al., 2022; Xie et al., 2023; Jia et al., 2023; Wang et al., 2023b; Shaoyong et al., 2023; Wang et al., 2023a; Wang et al., 2023c; Zeng et al., 2024; Sun et al., 2024). However, one study noted a decrease in Il1b mRNA expression (Sun et al., 2021). Regarding the anti-inflammatory cytokine IL-10, the data were contradictory: some studies demonstrated a decrease in its level (Wen et al., 2022; Jia et al., 2023), while others report—an increase in its mRNA expression (Sun et al., 2021; Choi et al., 2021a). There were isolated data on the IFNγ level increase and the mRNA Ifng, Tgfb, Il8, Il17a, Il22 expression when exposed to MP (Sun et al., 2021; Choi et al., 2021a; Liu et al., 2022a; Djouina et al., 2022; Luo et al., 2022; Zeng et al., 2024).

Several studies investigated MP effect on intracellular signaling pathways. According to the study of Choi et al. (2021a) results MP activated the ASC-inflammasome pathway: in the colon, the level of NLRP3 (NLR family pyrin domain containing 3), ASC (apoptosis-associated speck like protein containing a CARD) and the ratio of uncoupled caspase 1 to the total level of caspase 1 increased. The authors also demonstrated pro-inflammatory NF-κB pathway activation: in the colon, the level of phosphorylated NF-κB and phosphorylated IκB-α increased, as well as Nfκb mRNA expression. Choi et al. (2021b) revealed MP effect on the muscarinic acetylcholine receptor signaling pathway: the level of mAChR M2 and mAChR M3 receptors decreased, but Gα level increased, as well as PKC (protein kinase C) phosphorylation and PI3K (phosphoinasilitol 3 kinase) phosphorylation, which were mAChR downstream signaling molecules. The MAPK/NF-κB signaling pathway also changed: the level of ERK (extracellular-regulated kinase) phosphorylation, p38, NF-κB, and IκB increased. According to Jia et al. (2023) MP activated the TLR4/NF-κB signaling pathway: the TLR4, p50, and phosphorylated p65 levels increased, and the IκB level decreased. According to Li et al. (2020), MP consumption activated the pro-inflammatory pathway via TLR4: the level of TLR4, AP-1 (Activator protein 1) and IRF5 (Interferon regulatory factor 5) in the colon increased. The data from Sun et al. (2021), on the contrary, indicated the TLR4 signaling pathway suppression: Erk1 and Nfκb mRNA expression decreased, and Tlr4 and MyD88 mRNA expression did not change. The authors also assessed the expression of short-chain fatty acid receptors and did not detect changes in the Gpr41 (Ffar3) and Gpr43 (Ffar2) mRNA expression. In the work of Xie et al. (2023) activation of the Notch signaling pathway was observed: mRNA expression of Dll1, Dll4, Jag1 and Hes1 increased. Rawle et al. (2022) used RNA-seq to demonstrate that MP consumption caused pro-inflammatory signals moderate activation, sirtuin signaling pathway suppression, stimulated ribosomal and translational activity, and caused changes in the genes expression associated with the cytoskeleton and interaction with the extracellular matrix.

In 19 of the 34 articles analyzed, gut microbiota was studied using feces 16S rRNA or intestinal content sequencing. Only Rawle et al. (2022) concluded that MP consumption did not have a significant effect on the gut microbiome. The other studies revealed changes in the intestine microbiota composition during MP consumption (Li et al., 2020; Sun et al., 2021; Sun et al., 2024; Zhao et al., 2022; Shi et al., 2022; Chen et al., 2022c; Chen et al., 2022b; Djouina et al., 2022; Wen et al., 2022; Liu et al., 2022b; Xie et al., 2022; Wang et al., 2023a; Wang et al., 2023c; Lv et al., 2023; Yu et al., 2024; Zha et al., 2024; Liang et al., 2024; Jiang et al., 2024). Data on the microbiota diversity impact were contradictory: there were indications of the increase (Li et al., 2020; Liu et al., 2022b; Xie et al., 2022; Yu et al., 2024), and of the decrease (Shi et al., 2022; Liu et al., 2022a; Wen et al., 2022; Wang et al., 2023a; Wang et al., 2023c; Sun et al., 2024), and of zero changes (Sun et al., 2021; Chen et al., 2022c; Zha et al., 2024). There were little data on the increase in the gut microbiota abundance and richness (Li et al., 2020; Chen et al., 2022c; Liu et al., 2022b; Xie et al., 2022). Zha et al. (2024) reported an increase in fungal richness. The predominant bacterial types in mice gut microbiota were Firmicutes and Bacteroidetes. According to the obtained data, the Firmicutes/Bacteroidetes ratio increased when exposed to MP (Li et al., 2020; Zhao et al., 2022; Shi et al., 2022; Wen et al., 2022; Wang et al., 2023c), according to others, it decreased (Sun et al., 2021; Liu et al., 2022b; Xie et al., 2022; Wang et al., 2023a; Yu et al., 2024), and according to Sun et al. (2024) it did not change. The studies revealed many changes in the content of microorganism’s families and genera under the MP influence, but they varied greatly depending on the type of plastic and the timing of the experiment, which made it not possible to compare data from different studies. It should be stated the important observation of Liu et al. (2022b): in their work, under MP influence, the probiotic bacteria ratio decreased and the pathogenic bacteria ratio increased. Thus, MP can cause disturbances in the intestinal microflora composition.

A number of mouse colon and feces samples metabolomic studies were conducted. All data indicated that MP affected the metabolism of intestinal microflora and cells, changed the metabolism of amino acids, carbohydrates, lipids, vitamins and cofactors. However, the severity and direction of specific changes varied significantly in different studies (Sun et al., 2021; Shi et al., 2022; Chen et al., 2022c; Chen et al., 2022b; Liu et al., 2022a; Wen et al., 2022; Wang et al., 2023a; Wang et al., 2023c; Zha et al., 2024; Jiang et al., 2024; Garcia et al., 2024).

Table 2 MP influence on colitis course.

Reference	Experimental conditions	MP+colitis group versus colitis without MP group	
Zolotova et al. (2023)	Animals: adult male C57BL/6 mice
MP: PS, size 5 μ m, dose 2.3 mg/kg/day (suspension in drinking bowls 10 mg/ml), 1–42 d.e.,
Colitis: 1% DSS 36–40 d.e.
End of experiment: 7 day from the colitis start	↑ Colon ulcers and inflammation prevalence
↓ Neutral mucins in GC (↓ PAS-reaction intensity in GC)	
Xie et al. (2023)	Animals: adult male C57BL/6 mice
MP: PS, size 5 μ m, dose 0.01 mg/kg/day (suspension in drinking bowls 100 μ g/l), 1–42 d.e.
Colitis: 3% DSS 36–42 d.e.
End of experiment: 7 day from the colitis start	↓ Colon length
↑ Diarrhea and bloody stool 1.5x scores on the 3th day of colitis.
↑ Cecum and colon inflammation severity at the macro level
↑ Colon histological score
↑ LPS level in blood
↑ TNF-α, IL-1β and IL-6 levels and mRNA of Il1b and Il6 expression levels in colon
↓ Body weight
↑Il1b and Il6 mRNA expression in the liver
↑ Liver histological damage	
Zheng et al. (2021)	Animals: adult male C57 mice
MP: PS, size 5 μ m, dose 0.1 mg/kg/day, 1–28 d.e.
Colitis: 3% DSS 7 days.
End of experiment: 7 day from the colitis start	↑Intestinal permeability (Serum FITC-dextran content)
↑ IL-1β, TNF-α and IFN-γ levels in blood
↑ Liver inflammatory infiltration
↑ Fat vacuoles in the liver
↑ MDA and PPAR-γ (peroxisome proliferator-activated receptor levels
Compared to the colitis group, the MP+colitis group exhibited distinct alterations in various metabolic pathways, including alanine, aspartate, and glutamate metabolism, phenylalanine metabolism, D-glutamine and D-glutamine metabolism, and others.	
Luo et al. (2022)	1. Animals: adult male C57BL/6J mice
MP: PS, size 5 μ m, dose 0.02 and 0.2 mg/kg/day (gastric tube), 1–63 d.e.
Colitis: 2% DSS 1–7 d.e.
End of experiment: 21 day from the colitis start
2. Animals: adult male C57BL/6J mice
MP: PS, size 5 μ m, dose 0,02 mg/kg/day (gastric tube), 1–63 d.e.
Colitis: 1–2% DSS 1–7, 22–28, 43–49 d.e.
End of experiment: 63 day from the colitis start	21st day of colitis:
↓ Colon length (high dose)
↑ Colon tissue damage
↑ LPS blood level (both doses)
↓ GC volume fraction (AB-PAS stain)
↑ Genes associated with inflammation and immune response, including Tgfb (low dose), Cox2, Il17a (both dose), Il22 (high dose) mRNA expression in colon
↑ TNF-α and IL-10 levels in blood
↑ Liver histological damage
63rd day of colitis:
↑ colon tissue damage
↑ LPS blood level
↑ Genes associated with inflammation and immune response, including Il1b, iNOS, Cox2 mRNA expression level
No changes: IL-β, IL-6, TNF-α and IL-10 blood levels
↑ Liver histological damage	
Ma et al. (2023)	Animals: adult male C57BL/ 6J mice
MP: PS, size 0,1 μ m, dose 1, 5 and 25 mg/kg/day (gastric tube), 6–33 d.e.
Colitis: 2,5% DSS 1–5 d.e., 2% DSS 17–21 and 29–33 d.e.
End of experiment: 33 day from the colitis start	↓ Colon length (all doses)
↑ Colon pathological score (high dose)
↑Tnfa mRNA expression (high dose) and
↓Il10 mRNA expression (all doses) in colon
↑ MAPK signaling pathway activity in colon:
↑ Erk1/2 phosphorylation level (all doses), JNK and p38 (high dose)
↑ Relative liver mass (high dose)
↑ Liver pathological score (all doses)
↑ Cholesterol and blood glucose (high dose) levels
↑ Oxidative stress in the liver:
↑ MDA (all doses) and
↓ SOD, GSH and T-AOC (high dose)
Lipid metabolism in the liver changes (high dose)	
Liu et al. (2022a)	Animals: adult male C57BL/6 mice
MP: PS, size 5 μ m, dose 0.1 mg/kg/day, 8–35 d.e.
Colitis: 3% DSS 1–7 d.e.
End of experiment: 35 day from the colitis start	↑ Colon histopathological damage
Compared with the colitis group, the MP+ colitis group had different colon microbiome and metabolism	
Schwarzfischer et al. (2022)	Animals: female C57BL/6 mice immediately after weaning
MP: PS, size 0.05 1 μ m, dose 10 mg/kg/day (suspension in drinking bowls 0.05 mg/ml), 1–178 d.e.
Colitis: 1,5% DSS 169–175 d.e.
End of experiment: 10 day from the colitis start
2. Animals: female mice C57BL/6 immediately after weaning
MP: PS, size 0.05 1 μ m, dose 10 mg/kg/day (suspension in drinking bowls 0,05 mg/ml), 1–166 d.e.
Colitis: 1–1,5% DSS 85-91, 102–108, 119–125, 136–142 d.e.
End of experiment: 82 day from the colitis start	PS accumulated in the small intestine and organs distant from the gastrointestinal tract, but PS intake did not affect intestinal health and did not worsen colitis. It accumulated in the small intestine, mesenteric lymph nodes, spleen, and liver, but was not detected in the colon.
10th day of colitis:
MP exposure did not worsen acute colitis:
- weight loss was similar in all DSS-treated groups, regardless of MP particle exposure
- according to endoscopy and histology, the inflammation severity was comparable in all DSS groups, regardless of plastic treatment
- there were no differences in spleen weight and colon length between the DSS-treated groups
- plastic exposure did not alter the proinflammatory cytokines or barrier molecules (Tnfa, Ifng, Il12b, Cdh1, Cldn1 and 2, Ocln) expression
82nd day of colitis:
MP exposure did not worsen chronic colitis:
- similar weight loss in all DSS groups and equivalent recovery time
- no differences between the DSS groups based on colonoscopy, histology, spleen weight, and colon length were observed	
Notes.

MP microplastic

DSS dextran sulfate sodium

PS polystyrene

d.e. day of the experiment

GC goblet cells

MP influence on the acute and chronic experimental colitis

The experimental conditions varied in the seven identified publications (Table 2). The experiments were performed on C57BL/6, C57BL/6J or C57 mice, usually males were used, and in only one study were used females. All studies used PS microparticles, size 5 µm in five studies, 0.1 µm in one study, and 0.05 and 1 µm in another study. MP ranged from 0.01 to 25 mg/kg/day. MP exposure duration ranged from 3 weeks to 6 months, mainly 4–6 weeks. Acute colitis was induced by replacing drinking water with 1–3% dextran sulfate sodium (DSS) solution for 5–7 days. To induce chronic colitis, DSS cycles with recovery with drinking water for 1–2 weeks were 3–4 times repeated. The colitis induction in five experiments was initiated 5–24 weeks after preliminary MP consumption, in two experiments DSS and MP exposure started simultaneously, and in another two experiments MP started after the DSS course. Animals were withdrawn from the experiment 7, 10 or 21 days after the start of colitis induction (acute and subacute colitis) or 33, 35, 63 or 82 days after the start of colitis induction (chronic colitis) (Fig. 3).

Data from six studies indicated that MP consumption leads to a more pronounced course of both acute and chronic experimental colitis (Zheng et al., 2021; Liu et al., 2022a; Luo et al., 2022; Xie et al., 2023; Zolotova et al., 2023; Ma et al., 2023). On the 3rd day of colitis, according to Xie et al. (2023) diarrhea and bloody stool scores were 1.5 times higher with MP consumption. On the 7th day of DSS colitis development, according to Zolotova et al. (2023), mice that received MP exhibited a higher prevalence of ulcers and inflammatory infiltration in their colon compared to mice that did not receive MP. Additionally, they observed lower levels of neutral mucins in goblet cells. According to Xie et al. (2023) at the same colitis time points, MP consumption led to a more pronounced colon shortening, a higher inflammation severity in the cecum and colon at the macroscopic level and histological score of the colon, an increase in the level of proinflammatory cytokines TNF-α, IL-1β and IL-6 and Il1b and Il6 mRNA in the colon and the level of lipopolysaccharide in the blood. Authors also noted more pronounced inflammatory infiltration and dystrophy in the liver and higher Il1β and Il6 mRNA expression in it on the 7th day. Zheng et al. (2021) reported the increased intestinal permeability on the 7th day of colitis under MP influence, as well as higher levels of proinflammatory cytokines IL-1β, TNF-α and IFN-γ in blood and more severe liver damage. In the liver, a more pronounced inflammatory infiltration, a greater number of fat vacuoles, higher levels of MDA and PPAR-γ (peroxisome proliferator-activated receptor γ), alterations in the metabolism of alanine, aspartate, and glutamate, phenylalanine, d-glutamine, and d-glutamine metabolism, and other changes were observed.

Figure 3 Modeling the effect of microplastics (MP) on the development of DSS-induced colitis.

Note. Zolotova et al., 2023; Xie et al., 2023; Zheng et al., 2021; Schwarzfischer et al., 2022; Luo et al., 2022; Ma et al., 2023; Liu et al., 2022a.

On the 21st day of colitis, mice consuming MP demonstrated more pronounced colon shortening, more pronounced histological changes in the colon, a lower goblet cells volume fraction and higher intestinal permeability (higher LPS level in the blood). The colon demonstrated higher mRNA expression of genes associated with inflammation and immune response, including Tgfb, Cox2, Il17a, Il22, and the blood demonstrated higher TNF-α and IL-10 levels. In addition, more pronounced histological changes were detected in the liver (Luo et al., 2022).

On the 33rd day of colitis, under MP influence in the colon, there was also a more pronounced colon shortening, a higher Pathological score, higher proinflammatory cytokine Tnfa mRNA expression, and lower anti-inflammatory cytokine Il10 mRNA, as well as higher activity of the MAPK signaling pathway: higher levels of phosphorylation of Erk1/2, JNK and p38. In the blood were revealed higher glucose and cholesterol levels. The relative weight of the liver was higher, it had a higher Pathological score, oxidative stress (higher MDA level and lower SOD, GSH and T-AOC indicators), and changes in lipid metabolism were observed (Ma et al., 2023). On the 35th day of colitis in mice consuming MP, histological examination of the colon demonstrated more pronounced pathological changes: severe epithelial damage and vacuolation of the colonic mucosa, with disordered goblet cell structure, and a large number of inflammatory cells gathered near the intestinal mucosa. The authors also stated that differences in the intestinal microflora composition and the intestinal metabolome were detected between the groups of animals with colitis that received and did not receive MP (Liu et al., 2022a).

On 63rd day of colitis with MP consumption, there was a tendency for the histological score to increase in the colon in comparison to colitis without MP, and the LPS level in the blood was higher. In the colon, there was a higher mRNA expression of genes associated with inflammation and immune response, including Il1b, iNOS, Cox2, but the blood level of cytokines IL-β, IL-6, TNF-α and IL-10 did not differ between the groups of animals with colitis receiving and not receiving MP (Luo et al., 2022).

However, according to Schwarzfischer et al. (2022) MP did not aggravate the course of either acute (10th day) or chronic (84th day) colitis. The authors found accumulation of MP particles in the small intestine, mesenteric lymph nodes, spleen and liver, but did not detect it in the colon. According to endoscopy and qualitative histological examination, the severity of inflammation in the groups of animals with colitis that received and did not receive MP was comparable. There were also no differences in body weight, spleen weight, colon length, expression of proinflammatory cytokines or barrier molecules (Tnfα, Ifng, Il12b, Cdh1, Cldn1, Cldn2, Ocln).

Conclusions

In healthy mice, MP can cause colon damage: oxidative stress, increased permeability, immune cell infiltration of the mucosa, increased proinflammatory cytokines production, decreased goblet cell number and mucus production. MP consumption led to changes in proliferation, apoptosis and differentiation of epithelial cells, expression of tight junction and glycocalyx components, membrane transport, intracellular signaling pathways, metabolome and composition of intestinal microflora. In acute and chronic experimental colitis, most data suggest that MP exacerbates the pathological process, leading to a more severe course. According to these data, MP should be considered as one of the factors contributing to colitis development in humans. However, the type of plastic, particle size, dosage and exposure time in studies varied significantly, which makes it difficult to compare the obtained results. Therefore, further research is essential to evaluate the potential health risks associated with MP consumption.

Abbreviations and symbols

AB-PAS Alcian Blue and Periodic acid–Schiff staining

AP-1 Activator Protein 1

Aqp Aquaporin

ASC Apoptosis-associated Speck like protein containing a CARD

Bax BCL2-Associated X Protein, apoptosis regulator

Bcl2 B-Cell CLL/Lymphoma 2, apoptosis regulator

Bmi1 B Lymphoma Mo-MLV Insertion Region 1 homolog

CAT Catalase

CD11b Cluster of Differentiation molecule 11B

CFTR Cystic Fibrosis Transmembrane conductance Regulator

Chga Chromogranin A

Clc2 Chloride channel 2

Cldn Claudin

Cmyc Cellular homolog of the retroviral v-Myc oncogene

Cox2 Cyclooxygenase-2

DAO Diamine oxidase

D-Lac D-Lactate

D-LDH D-Lactate dehydrogenase

Dll Delta Like Canonical Notch Ligand

DSS Dextran Sodium Sulfate

Ecad E-cadherin

ERK Extracellular-Regulated Kinase

F11r F11 Receptor, junctional adhesion molecule A

Gpr G-protein-coupled receptors

GPx Glutathione peroxidase

GSH Glutathione

GSSG Oxidized Glutathione

Hes1 Hairy and enhancer of split homolog-1

IBD Inflammatory bowel diseases

IFABP Intestinal Fatty Acid-Binding Protein

IFNγ (Ifng) Interferon gamma

IL (Il) Interleukin

iNOS Inducible Nitric Oxide Synthase

IRF5 Interferon Regulatory Factor 5

IκB Inhibitor of Nuclear Factor Kappa-B

Jag1 Jagged Canonical Notch Ligand 1

JNK c-Jun N-terminal kinases

Klf4 Kruppel-like factor 4

LDPE Low-Density Polyethylene

Lgr5 Leucine-rich repeat-containing G-protein coupled receptor 5

LPS Lipopolysaccharide

Ly6C Lymphocyte Antigen 6C

mAChR Muscarinic acetylcholine receptors

MAPK Mitogen-Activated Protein Kinase

MDA Malondialdehyde

MP Microplastic

Muc Mucins

MyD88 Myeloid differentiation primary response 88

NF-κB Nuclear Factor Kappa-B

NKCC1 Na-K-2Cl cotransporter 1

NLRP3 NLR family pyrin domain containing 3

Ocln Occludin

Olfm4 Olfactomedin 4

Ox-LDPE Oxidized Low-Density Polyethylene

p38 a protein that belongs to the mitogen-activated protein kinase (MAPK) family

p50 a protein that belongs to the Y-box binding transcription factor family

p65 Transcription factor p65 also known as nuclear factor NF-kappa-B p65 subunit

PAS-reaction Periodic acid–Schiff reaction

PCNA (Pcna) Proliferating Cell Nuclear Antigen

PE Polyethylene

PET Polyethylene Terephthalate

PI3K Phosphoinasilitol 3 Kinase

PKC Protein Kinase C

PLA Polylactic Acid

PLGA Poly Lactic-co-Glycolic Acid

p-NF-κB, p-I κB-α phosphorylated proteins

POD Peroxidase

PP Polypropylene

PPAR-γ Peroxisome Proliferator-Activated Receptor γ

PS Polystyrene

PVH Polyvinyl Chloride

Retnlb Resistin-like beta

ROS Reactive Oxygen Species

SLC26A6 Solute Carrier Family 26 member 6

SOD Superoxide Dismutase

T-AOC Total Antioxidant Capacity

Tff3 Trefoil factor 3

TGFβ(Tgfb) Transforming growth factor beta

TLR Toll-like receptors

TNBS 2,4,6-Trinitrobenzenesulfonic Acid

TNFα (Tnfa) Tumor necrosis factor α

TUNEL Terminal deoxynucleotidyl transferase dUTP Nick End Labeling

Vil1 Villin 1

ZO-1 Zonula Occludens-1

Additional Information and Declarations

Competing Interests

Author Contributions

Data Availability

The authors declare there are no competing interests.

Natalia Zolotova conceived and designed the experiments, performed the experiments, analyzed the data, prepared figures and/or tables, authored or reviewed drafts of the article, and approved the final draft.

Dzhuliia Dzhalilova performed the experiments, authored or reviewed drafts of the article, and approved the final draft.

Ivan Tsvetkov performed the experiments, prepared figures and/or tables, authored or reviewed drafts of the article, and approved the final draft.

Maria Silina performed the experiments, analyzed the data, authored or reviewed drafts of the article, and approved the final draft.

Nikolai Fokichev analyzed the data, prepared figures and/or tables, authored or reviewed drafts of the article, and approved the final draft.

Olga Makarova conceived and designed the experiments, analyzed the data, authored or reviewed drafts of the article, and approved the final draft.

The following information was supplied regarding data availability:

This is a literature review.

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
