# Peer review of "Microplastic effects on mouse colon in normal and colitis conditions: A literature review"

_PeerJ, doi:10.7717/peerj.18880_

## Round 0.1 · original submission · Major Revisions

Two experts have assessed your manuscript and identified a number of issues that make the manuscript unacceptable in its present form. The most important aspects are that (1) the grammar and spelling must be substantially improved. You also should seek support from a language editing service. (2) The citation practice should be updated as indicated by reviewer 1.

I hope that their criticism will allow you to carry out a substantial revision of the manuscript, which is a precondition for its acceptance.

Reviewer 1 ·

Basic reporting

The manuscript contains quite some typos, misspellings, Cyrillic letters in the table and parts are not well formulated in English or sound rather unscientific. Thus, the manuscript should be thoroughly revised with spelling and grammar checks and also a language editing service would be highly recommended to improve the readability. Also the authors should focus on being consistent with the use of words/terms and how things are written in the table (e.g. ICR mice vs mice ICR, start with big/small capital letters, use of “.” sometimes at the end).
The references, especially in the introduction, are sparsely given and mostly only at the end of each paragraph. Also mostly reviews are cited for the introduction and not the original article/source. The authors are thus addressed to cite every reference properly with the original reference and directly in/after the sentence (especially when mentioning statistical data or any specific numbers).
The article is of broad interest for the specific field of microplastic research, however the last big review of the same topic was published only in 03/2023 in Chemosphere. Aside from an updated list of publications no new aspects are discussed.

Experimental design

The authors did a literature research of open access original articles investigating the effects of microplastics on the colon of healthy mice and in colitis (34 +7 articles). Findings of these studies were listed in two tables and common findings or contradicting findings were summarized for various aspects in the text. The review is organized in a logical way, however the results part seems to be rather an accumulation of the various research findings with excessive citation of the findings from the various papers, and does not give a well readable overview/summary about the potential MP induced effects.

Validity of the findings

The authors state that “The purpose of our review is to summarize and systematize the data on the MP effect on mice colon under normal conditions and during colitis in order to assess the role of MP in the development of intestinal diseases”. However, this statement seems a little bit far fetched, as “intestinal diseases (or “inflammatory bowel diseases” as it is also often stated in the manuscript) comprise multiple different diseases in the small and large intestine, and in this article the focus is only on the large intestine and colitis. Although most diseases summarized under the term IBD have a chronic inflammation of intestinal tissue in common, the authors even miss to correlate the findings from colon tissue (in healthy mice) and colitis bearing mice to the other parts of the intestine or to any other IBD disease. Thus, the authors should not only limit their search to large intestine and colitis, but also investigate MP effects in other parts of the intestine and IBD diseases. Furthermore, the findings for colon and colitis should be also discussed in the context of other IBD diseases.

Additional comments

Only open access articles were analysed, thus making the analysis maybe a little bit biased.
For certain statements only one reference is given, depicting just one specific aspect and not multiple as it should be in the context of a review.
e.g. for the estimated uptake of MPs in humans only the publication from Senthirajah et al. 2021 is cited (5g/week), which gives rather unrealistically high doses. This is currently in the microplastic community heavily discussed. Thus, the authors are advised to not only cite this one article, but also include some of the other studies that majorly give a lower dose that might be a more physiological realistic dose estimate, or at least cite more recent publications.
The group has a specific publishing background in this journal:
“Harmful effects of microplastic pollution on animal health: a literature review” 2022 “Murine models of colorectal cancer: the azoxymethane (AOM)/dextran sulfate sodium (DSS) model of colitis-associated cancer”

·

Basic reporting

• Clear, unambiguous, professional English language used.
• Intro & background to show context. Literature well referenced & relevant.
• Structure conforms to PeerJ standards, discipline norm, or improved for clarity.
the review of broad and cross-disciplinary interest and is within the scope of the journal.
The current review has different viewpoint
• Introduction adequately introduces the subject and makes audience and motivation clear.

Experimental design

• Article content is within the Aims and Scope of the journal.
• Rigorous investigation performed to a high technical & ethical standard.

• Methods described with sufficient detail & information to replicate.

• The Survey Methodology is consistent with a comprehensive, unbiased coverage of the subject. But need Draw PRISMA flow chart to explain how to retired final papers

• The review is organized logically into coherent paragraphs/subsections.

Validity of the findings

• Impact and novelty is not assessed. Meaningful replication encouraged where rationale & benefit to literature is clearly stated.

• Conclusions are well stated, linked to original research question & limited to supporting results.

• There is a well-developed and supported argument that meets the goals set out in the Introduction

• The Conclusion identified unresolved questions / gaps / future directions.

Additional comments

In this paper the authors reviewed microplastic effects on mouse colon in normal and colitis conditions. The paper is important and have the novelty in the point of continuously growing awareness of the toxicity of microplastics which are now present everywhere in the environment. Overall, the study is interesting and addresses a relevant research topic that may interest PearJ readers. However, I recommend that some adjustments/improvements be made before publishing the manuscript.
Specific comments/suggestions:
1. Line 27: Diseases not desiases.
2. Line 29: Replace (animals with colitis) by mice or rephrase this sentence.
3. Line 61: remove the sentence (MP entered into the organism with water and food accumulated in organs).
4. Your introduction needs more details. I suggest that you add paragraph about toxicokinetic of microplastics.
5. Line 91-95: Repeated sentences as in abstract … change or improve the sentence.
6. Survey Methodology: Should draw PRISMA flow chart to explain how to the authors retired the final papers.
7.Line 124-125: animal sex is in 33 papers not in 34 papers, there was missed paper.
8. Line 127: type of plastics, 76.5% is equal to how many studies? Write number of them.
9. Line 148: It is better to replace (increase distance between crypts) with loss of crypts parallelism.
10. Line 157: It is better to replace (mucous and muscular membrane thickness) with mucosa and muscularis externa layers.
11. Line 158: It is better to replace (decease in the superficial epithelial height) with decease in the superficial enterocytes height or superficial absorptive cells.
12. Line 159: Write histopathological scoring or pathological score instead of histology score
13. Line 164: Remove word barrier from (colon epithelial barrier), instead of it …. colon epithelium.
14. Line 174:
(Choi et al 2021b) stated that decreased number and volume fraction of goblet cells, but in line 179 the same reference said there was ultrastructure increased the number of mucin droplets in goblet cells….. Revise these sentences.
15. Line 199: Replace endocrine with enteroendocrine cells.
16. Line 374: Revise is it colon shortening or intestinal ?... to exclude small intestine
17. Line 381: Replace cilia with microvilli
18. Add any abbreviation in the footnote of both tables.
19. In Fig.1 , revise time of exposure of MP in all 7 papers.
20. Finally, the English language should be improved to ensure that an international audience can clearly understand your text. I suggest you have a colleague who is proficient in English and familiar with the subject matter review your manuscript, or contact a professional editing service.

---

## Round 0.2 · Minor Revisions

Thank you for the revision of the MS. We are almost there. However, as pointed out by reviewer 1, there are still a number of issues to be solved before the MS can be accepted, including the improvement of the language. The reviewer proposes to use a professional English language editing service or software like Grammarly and I support this proposal fully. Also the other proposals of reviewer 1 should be implemented in the final revision.

I look foward receiving the revised manuscript.

Reviewer 1 ·

Basic reporting

The authors improved the language quality of the manuscript, however it still contains multiple typos, misspelling and bad english wording.

e.g. “ ore” instead of or, µ m (with space in between), 2 brackets opening but only one closing,
“The number of MP particles detected in tumor tissue was higher than in tumor-free colon areas of the same patients and then in patients with no colorectal cancer.”

It is still highly recommend to use professional English language editing service or the use of freeware like Grammarly to improve the readability of the manuscript.

Experimental design

no additional comments

Validity of the findings

The authors should stick to the term “colitis”, as this is the kind of disease that they investigated and described in their review. The more general term “inflammatory bowel disease” that is also sometimes used at certain areas instead of colitis is misleading and of incorrect use here. The manuscript is just focusing on colitis and no other IBD diseases like Crohn’s disease. Thus, a generalization of only colitis for IBD is not correct and should be avoided. (e.g. "According to these data, MP should be considered as one of the factors contributing to IBD development in humans." > replace IBD with colitis )
We recommend the authors to carefully revise the manuscript and use only the term colitis and to avoid these incorrect generalizations under the term IBD unless they provide data on Crohn's disease as well.

A statement in the conclusion, which parameters were the most robust to assess in terms of MP effect on colon/colitis, would be recommended

Additional comments

The addition of other citations in the context of human microplastic exposure in the introduction paragraph is appreciated. However, the bulletpoint list, doesn’t fit the flow of the text.

·

Basic reporting

- The review is within the scope of the journal
- The review has a different point of view
- The introduction adequately introduce the subject

Experimental design

The survey methodology is sufficient

Validity of the findings

There is a well developed and supported argument that meet the goals set out in the introduction

---

## Round 0.3 · accepted · Accept

Thank you for the thorough revision of the manuscript. I hereby certify that you have adequately taken into account the reviewers' comments and improved the manuscript accordingly. Based on my assessment as an Academic Editor, your manuscript is now ready for publication.